# Network Traffic Modeling in a Wi-Fi System with Intelligent Soil Moisture Sensors (WSN) Using IoT Applications for Potato Crops and ARIMA and SARIMA Time Series

**Alfonso José López Rivero [1], Carlos Andrés Martínez Alayón [2,*], Roberto Ferro [3], Daniel Hernández de la Iglesia [1] and Vidal Alonso Secades [1]**

[1] Faculty of Informatics, Universidad Pontificia de Salamanca, C/Compañía 5, 37002 Salamanca, Spain; ajlopezri@upsa.es (A.J.L.R.); dhernandezde@upsa.es (D.H.d.l.I.); valonsose@upsa.es (V.A.S.)

[2] Faculty of Engineering, Universidad ECCI, Cra. 49 # 49-20, Bogotá 111311, D.C., Colombia

[3] Faculty of Engineering, Universidad Distrital Francisco José de Caldas, Cra. 7 # 40B-53, Bogotá 111311, D.C., Colombia; rferro@udistrital.edu.co

* Correspondence: cmartineza@ecci.edu.co

**Abstract:** This article presents the results obtained by analyzing the data traffic that originated in a system with intelligent soil moisture sensors (Wireless Sensor Network—WSN) that transmit through a wireless network. This study sought to integrate smart agriculture and IoT (Internet of Things) applications in potato crops in various rural settings. Using these measurements, the data analysis was performed through the ARIMA (autoregressive integrated moving average model) and SARIMA (seasonal autoregressive integrated moving average model) time series following the Box–Jenkins methodology. GRETL (Gnu Regression, Econometrics and Time-series Library) free software was used to generate a teletraffic behavior prediction model in a larger-scale implementation. The main objective was the creation of a model that allows an analysis and simulation about the behavior of the main performance parameters that a medium-scale WSN system would have for the monitoring of a crop. Thanks to this analysis, it will be possible to determine the technical characteristics that a sensor deployment should have in a specific area and for a specific crop.

**Keywords:** Smart rural; IoT; sensors; ARIMA; SARIMA; teletraffic; precision agriculture; WSN, LPWAN

## 1. Introduction

Due to the current boom in data networks and especially those that support the growth of the IoT (Internet of Things) and rural Smart (smart agriculture), it is vitally important to develop models that allow us to better understand and represent their operation. Among the different options that exist to model some types of traffic, the statistical models stand out, since they better represent stochastic processes using probabilistic tools and allow treating the noncontinuity of the data traffic generated in the networks, represented by packets, bursts, and session [1]. That is why time series and the methods to model them have begun to be used in traffic analysis, given their advantages to represent and predict processes that fluctuate randomly with a known time reference.

The LIDER research group, Laboratory for Research and Development in Electronics and Networks of the Francisco José de Caldas District University of the city of Bogotá (https://comunidad.udistrital.edu.co/grupolider), has a bank of projects in Smart Agriculture (Smart rural). It also has a line in IoT and smart cities, where several investigations are carried out focused on the optimization of processes in these areas, thanks to the use of resources and technological means.

The study of these issues gave rise to the creation of the SCISEN research seedbed (Smart Cities & Sensor Network), which, with the participation of students and teachers, has made significant progress in the area of smart agriculture in Bogotá, Colombia.

One of the projects developed is based on the analysis of crops, such as potatoes. This study is in the central-eastern region of Colombia and seeks to establish optimal environmental and climatic conditions, soil characteristics, and other variables. Using a set of technologies integrated into the concept of Smart Agriculture, it seeks to establish the most optimal way to work with telecommunications networks, traditional wireless, and LPWAN networks (low power wide area network). Also, together with software applications and other devices, such as sensors, development cards, etc., it seeks to contribute to the consolidation of agricultural productivity in the sector [2].

Also, it was considered as antecedent to this research the importance that, at a global level, the application of smart agriculture in rural sectors has taken to improve productivity and strengthen food security in several countries of the world. The processes of technology transfer to the agricultural sector, generally associated with terms such as smart agriculture and/or smart farming, are seen by various government entities, such as the Food and Agriculture Organization (FAO), as a key factor to effectively address the future trends [2]. There is a need for optimized and more environmentally sustainable agriculture to support the growth of the world population, avoid the harmful effects of climate change, and preserve water and natural resources.

Additionally, the IoT becomes an integrating and essential element for work in smart agriculture since it has important projections and applications in the immediate future. First of all, the agricultural sector has to be continuously monitored and controlled, which generates a large amount of data, for example, relevant environmental parameters for proper plant growth in a crop, which must effectively be collected, transferred, processed, and stored. In most cases, the deployment of data networks that generate connectivity in the field allows the exchange of information between nodes of an IoT architecture, and there the use of energy sources (for example, solar energy) to provide feeding to this scheme is mandatory.

In Colombia, an effort has been made to start the application of the concepts of smart agriculture and precision agriculture in coffee [2], sugar cane [3], and strawberry [4] crops, among other crops, in different regions (Llanos orientales, coffee axis, Valle del Cauca). However, one of the main difficulties is the lack of technological coverage and connectivity to the internet to be able to transmit the data acquired in rural sectors to the municipal capitals, where there are the points that are part of the Live Digital Plan implemented by the Ministry of ICT (Information and communication technologies) in Colombia and other government institutions in agreement with the National Development Plan 2010–2014. This void, which the government has not been able to solve in Colombia over several years and which currently has been evidenced more strongly by the effects of the COVID-19 pandemic, demonstrates the connectivity problems in a large percentage of rural Colombian territory, and constitutes the origin of the research that is presented in this document.

Specifically, a prototype of a network of nine smart soil moisture sensors, FC28, was implemented on the ESP8266 NodeMCU development platform that transmits under the 802.11n protocol (Wi-Fi) in the 2.4 GHz band. These signals were received by a LINKSYS WAP300N reference router that centralized the signals from the ESP8266 modules previously configured in station mode with addresses assigned by the router's DHCP (Dynamic Host Configuration Protocol) service. The assigned addresses were configured in the range between 192.168.0.10/24 to 192.168.0.30/24. A LAN (Local area network) was implemented with a gateway that was configured at address 192.168.0.1/24. This wireless LAN network, implemented in the building of the Faculty of Engineering in basement 3 where the facilities of the LIDER research group are located, has a coverage of approximately 90 m around the point where the main router is located. However, the tests were conducted with the sensors located at approximately 50 m around the router. Likewise, the computer where the Wireshark software was installed was connected to this same LAN network, which was used as a network sniffer to capture, filter, and analyze data packets sent in the wireless network, as described in [5]. With this, it was proposed to carry out

an analysis of data traffic in order to model and characterize the performance of the wireless channel to predict its behavior and apply it in a larger-scale implementation; more specifically, in a smart agriculture project that benefits the cultivation, for example, of potatoes in Colombia and Spain [6], where this case study was conducted. Not limited only to potatoes, other irrigated crops such as vegetables or beets can also benefit from the system.

The Linksys WAP300N router has a transmission capacity of up to 300 Mbps in the 2.4 GHz and 5.0 GHz bands. However, this speed by standards is reduced by headers, acknowledgments (in certain protocols), speed of the interfaces' host, processing capacities and memory, etc. Throughput for the interfaces can vary a lot. However, in this case it was not relevant, considering that the rest of the link limited the channel speed. The typical throughput for the signals transmitted in the study case was 80 Mbps. The antennas of the ESP8266 platforms were also limited by the standard used and offered a typical throughput of approximately 60 Mbps. Thus, the maximum values between points of the link, considering the speed and throughput specified by the standards, are shown in the Figure 1.

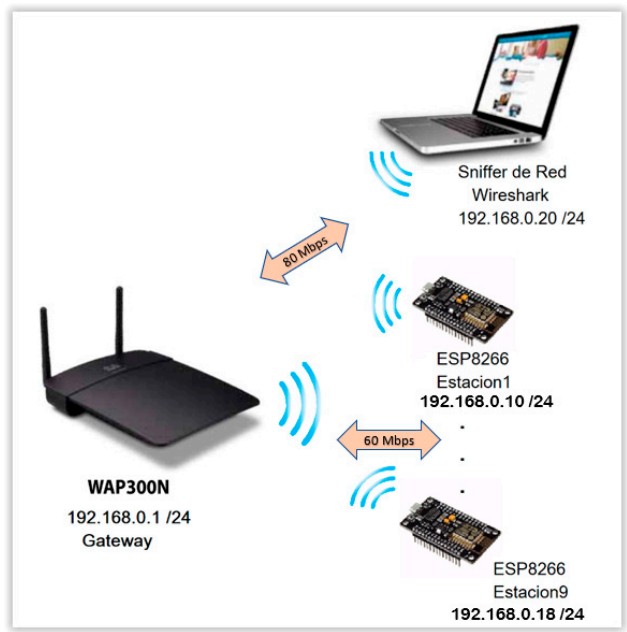

**Figure 1.** Assembly diagram of the Wireless Sensor Network (WSN).

In this article, the use of time series is proposed because the sequences of the samples of the observed variable were typically taken in successive instances of time and evenly spaced. To achieve this, 200 samples were taken from the humidity sensor that were read by a wireless device, the ESP8266, which transmitted them to a WiFi device. Once they were received on a computer, they were ordered for processing by means of a free software application that allows different models to be developed and data predictions (forecast) made. In this document, these predictions will be used to represent the traffic in the channel, given the temporal dependence that the data bursts present in a transmission.

Likewise, it is known that for network services based on wireless transmissions, packets directly affect the bandwidth used and the quality of service (QoS) in the channel. Therefore, the representation of data traffic measured in kbps (kilobits per second) through time series and its modeling by means of ARIMA and SARIMA was proposed. For this, a formal analysis of the bandwidth used during a data transmission in the sensor network was carried out. To obtain the models, use was made of the GRETL free software package [7], and the Box–Jenkins methodology [8] was followed. Thanks to the obtained models, it was possible to discover the behavior of the main performance parameters that a medium-scale WSN system would have for the monitoring of a crop. Thanks to this analysis, the technical characteristics that a sensor deployment should have, in a specific area and for a type of crop, were determined. The main parameters that the system allowed us to determine were

energy consumption, range, and coverage; supplied bandwidth; error rate; and data loss. In this way, deployment over large areas of land was more efficient and economical.

In accordance with the foregoing, in the following sections a brief exposition of the basic concepts for the analysis of the time series and the ARIMA and SARIMA processes will be made. Then the Box–Jenkins methodology will be used to adjust the models exposed in a sample of the data traffic in the channel and predictions of the series will be made with the parameters obtained. Additionally, the results will be compared to determine the suitability of each model and its ability to represent the behavior of the original series. Finally, the best results are presented as a formal description of the traffic used during the transmission.

## 2. State of the Art

In the current scientific literature, several studies were found that showed the results of the use of WSN and IoT devices applied to agriculture and environmental management. However, the analysis of the information was carried out with techniques different from the case study presented in this document. The study and prediction of the behavior of data traffic variables using ARIMA- and SARIMA-type time series were carried out by the same authors and scientists from other universities, oriented toward other technological applications. Among the most relevant writings that address similar topics that allow a construction of the state of the art, the following can be cited:

In [9], a solution based on an intelligent irrigation system is proposed that takes into account water requirements, soil moisture levels, environmental conditions, and air humidity level and allows the optimization of water use with minimal human intervention as a smart city app.

In [10], it is shown how the use of sensors and the Internet of Things (IoT) is key to taking global agriculture to a more productive and sustainable path. It also presents the opportunities that the agriculture and environment sectors have with the use of recent advances in IoT, WSN sensor networks, information, and communication technologies. As the number of interconnected devices continues to grow, this generates more high-volume data with multiple modalities and spatial and temporal variations. Intelligent processing and analysis of this big data are necessary to develop a higher level of knowledge base and insights that result in better decision making, forecasting, and reliable sensor management [11]. This document shows a comprehensive review of the application of different machine learning algorithms in the analysis of sensor data within the agricultural ecosystem. In addition, it analyzes a case study on an IoT-based, data-driven smart farm prototype as an integrated food, energy, and water (FEW) system.

Santos, Uélison, and others in [12] write about the challenges to guarantee food for all the inhabitants of the planet. To this end, they propose an alternative that consists of increasing agricultural production using the IoT to improve the capacity of the soil and the protection of environmental resources. This article presents a model called AgriPrediction, which combines a short- and medium-range wireless network with a technological system that anticipates possible crop failures and notifies the farmer to make decisions and corrective measures. To achieve this, AgriPrediction presents a framework whose components are based on both LoRa (Long Range) IoT technology and the ARIMA-type time series prediction model. Their results demonstrated the feasibility of using LoRa in rural locations, as well as providing the advantages of having a prediction system to observe problems related to soil moisture and temperature. All the tests in this study were carried out on an arugula crop where an improvement in productivity was obtained close to 18% compared to the crop that used traditional methods.

In the article by Balamurugan and Sivakami [13], the authors propose the implementation of a NGIF (next-generation integrated farming) using LoRa-IoT to improve productivity, produce better crops, and minimize labor through adequate monitoring of livestock health, soil health, air temperature, humidity, proper watering at the right time, and protecting crops from birds and animals. The simulation result shows that a hybrid Wi-Fi and Lora WAN network can support different IoT connectivity technologies, reduce complexity, and minimize delays in decision making to

improve productivity with lower costs in a rural area. All this is thanks to the application of integrated agriculture or smart agriculture.

In [14], the same authors of this document present the results obtained when representing the losses in a digital video transmission by means of ARIMA and SARIMA models, following the Box–Jenkins methodology and making use of the R programming language to estimate the coefficients.

Haviluddin and Nataniel Dengen in [15] present the results of their research by performing network traffic analysis and forecasting using SARIMA, NARX (Nonlinear Autoregressive Exogenous), and BPNN (back propagation neural network) models based on a short-term time series data set. The determination of these prediction models offers an alternative for researchers to obtain more accurate prediction results in the analysis of traffic in data networks.

In the article titled "Internet of Things (IoT) for Smart Precision Agriculture and Farming in Rural Areas" [16], its authors show how the use of the Internet of Things (IoT) generates a new dimension in smart agriculture. Likewise, it is shown that it is possible to use fog computing and a long-range WiFi network to connect agricultural bases located in rural areas in an efficient way.

Leni Symeonaki and others, in their study "A Context-Aware Middleware Cloud Approach for Integrating Precision Farming Facilities into the IoT toward Agriculture 4.0" [17], present in detail a hierarchical, layered structure according to which all the functional elements of the system are adapted to the context, while the context awareness operation is achieved in a cloud-based distributed middleware component that is the core of the entire decision support system (DSS). This document shows a dynamic integration of precision farming (PF) systems on the Internet of Things (IoT) applied in agricultural environments through wireless sensor and actuator networks (WSAN).

The paper presented by Khalid Haseeb [18] documents a research result where a WSN based on IoT was used to observe the condition of yields and automate the precision of agriculture using various sensors seeking to strengthen agricultural production with improvement of production yields through intelligent decisions and obtaining information on crops, plants, measurement of temperature, humidity, and irrigation systems.

The document "Design and Development of IoT-Based Framework for Indian Aquaculture" [19] presents the design and development of an Internet of Things (IoT)-based framework that measures water quality in aquaculture farms and provides alerts to farmers so that they can take the necessary precautions and save themselves from great losses in India.

Finally, Nallakaruppan, in his paper [20], shows an application of the Internet of Things and machine learning algorithms such as decision tree and time series analysis to forecast the weather more accurately over a long period of time.

Other writings that address a problem similar to the issue raised in this article but with less impact are "Autonomous Crop Irrigation System using Artificial Intelligence" [20], A Survey on the Role of IoT in Agriculture for the Implementation of Smart Farming [21], and "Real-time water quality monitoring through Internet of Things and ANOVA-based analysis: a case study on river Krishna" [22]. Some documents were also considered for the construction of the state of the art, which, if true, are not mentioned directly in this writing but were relevant to various phases of the project described here. For this reason, they are included in the references of this article.

## 3. Potential Significance

Potato cultivation, at present, has a great relevance worldwide [6,23,24]. It is the fourth most important crop, after rice, wheat, and corn [25]. Potatoes are second to none in the diet and livelihoods of millions of people around the world. World potato production has increased considerably in recent years, mainly due to increased cultivation in developing countries. The improvement of the varieties, the seeds used, and the methods of cultivation has led to higher yields. Furthermore, demand has been driven by changes in eating habits in several countries, which has led to increased consumption of potato-based products with a higher degree of industrial processing [24]. The area sown and production in developing countries in 2005 exceeded those of industrialized countries for the first

time. Currently, the main producer is China, with 71 million tonnes, which represent more than 20% of world production [6].

Potato cultivation is an important source of income for many farmers. In the Andean region in South America it is one of the few commercial crops produced by small farmers. In the tropical lands of Bangladesh and India it is cultivated commercially, mainly under irrigation, as a winter crop. The potato is especially popular with farmers in the mountainous areas of Vietnam, who benefit from favorable prices on the local market. They produce potato as a catch crop, rotating it with rice and corn. There, the income generated by potatoes is equal to that of rice and is double that of corn and sweet potato.

In potato cultivation, humidity management, the subject of this article, is of great importance for productivity at harvest and for determining the best times of the year for planting. An increase in temperature leads to greater plant transpiration, which increases the water demand of the plants. This will cause water stress in many of the drier producing areas, causing a decrease in yields. This effect will worsen with greater intensity because of changes in the distribution of rainfall. Yields will decline further where irrigation is not available, to the point where potato cultivation becomes nearly impossible. The expected decrease in yields in several countries, particularly in tropical and subtropical regions, will reach 20–30%. The night temperature has a crucial influence on the formation of starch in the tubers, with the ideal being 15 to 18 °C. When this temperature exceeds 22 °C, the humidity in the soil decreases and the development of the tubers is severely affected. However, climate change is expected to have a favorable effect on yields in growing areas located at higher altitudes. Climatic conditions for potato cultivation are improving because of rising temperatures in mountainous regions. This favor yields and leads to an expansion of production toward higher and higher latitude areas. In some regions, potatoes may be grown as a winter crop.

## 4. Theoretical Framework

### 4.1. Time Series

Time series are sequences of one-variable samples typically taken at successive and evenly spaced instances of time, which, in this document, will be used to represent the losses in the channel, given the temporal dependence that the data bursts present in a data transmission. These series are statistical tools used to explain (and in some cases predict) the value that the analyzed variable takes at a given moment of time [14].

### 4.2. Time Series Modeling

The time series alone are reduced to a simple temporal organization of samples that allow the description of some basic parameters of a stochastic process. What really represents a great contribution in the study of this type of series are the methodologies and mathematical models that have been developed around the description and prediction of random variables. In particular, the autoregressive models of moving averages and all those that have been derived from the combination of these two. Below is a brief description of some of the models worked on throughout the investigation. The notation used is the following:

- $Y_t$ is time series to be analyzed.
- $u_t$ is blank function with zero average and constant variance.
- $d\ y\ D$ is degrees of normal and seasonal differentiation;
- $\Phi_p(L)$ is polynomial of order p of the autoregressive component.
- $\Phi_P(L)$ is polynomial of order P of the autoregressive seasonal component.
- $\Theta_q$ is polynomial of order q of the component of moving averages.
- $\Theta_Q(L)$ is polynomial of order Q of the seasonal component of moving averages.
- $S$ is period of the function if it presents seasonality; and



- $\mu$ is average of the original undifferentiated function.

### 4.2.1. Autoregressive Model (AR)p

With this model, described by Equation (1), the current value of the series is expressed as a function of the values that it took in the p previous samples weighted by a factor $\varphi\_i$ and of a present random disturbance.

$$\Phi_p(L)Y_t = u_t \tag{1}$$

### 4.2.2. Moving Average Model (MA)p

Consider that the value of the stationary series moves around a mean value μ. Furthermore, it assumes that the displacement of μ in the present time t is caused by infinite disturbances that occurred in the past, weighted by a factor $\theta\_i$ that measures the influence of each one of them on the present value of the series. Its mathematical description is reflected in (2).

$$Y_t - \mu = \Theta_q(L)u_t \tag{2}$$

### 4.2.3. ARMA (Autoregressive moving average) Model

As shown in the Equation (3), this model represents a time series as a combination of the two previous models:

$$\Phi_p(L)Y_t = \Theta_q(L)u_t. \tag{3}$$

### 4.2.4. ARIMA Model

Some series must be differentiated to eliminate trends or changing variances and thus obtain stationary series. The ARIMA model refers to an ARMA model that has been applied to a differentiated series. Thus, we have the following representation for the model:

$$\Phi_p(L)(1-L)^d(Y_t - \mu) = \Theta_q(L)u_t. \tag{4}$$

### 4.2.5. SARIMA Model

The SARIMA model, autoregressive and integrated of the seasonal moving average, is based on the ARIMA model, with some of its coefficients at zero and additional components to integrate the seasonal behavior of the series in the model. The SARIMA model has the following notation:

$$\Phi_P(L)\Phi_p(L)(1-L^s)^D(1-L)^d(Y_t - \mu) = \Theta_Q(L)\Theta_q(L)u_t. \tag{5}$$

### *4.3. Methodology*

For the choice of the models described above the Box–Jenkins [26] methodology was used, which can be summarized in three steps:

- Identification and selection of the model (stationarity, seasonality, autoregressive components, and moving averages);
- Estimation of the coefficients that best fit the parameters chosen by means of computational algorithms; and
- Validation of the model obtained.

## 5. Development of the Models

Taking into account that the proposed analysis focuses on bandwidth without including additional channel performance indexes, such as latency, jitter, or others, the univariate time series were chosen for the representation of the data, taking into account that they allow analyzing the behavior of the

variable itself without trying to explain the factors that influence it. The steps that were followed in obtaining the ARIMA and SARIMA models are presented in detail below.

*5.1. Organization of the Data*

To make a good representation with time series, a time interval must be chosen that captures in some way a descriptive behavior for the pattern to be analyzed. In econometrics, for example, this problem generally depends on the periodicity with which the data are obtained (monthly, annual, quarterly indices, etc.). In data traffic, each transmitted packet carries headers that provide information in an almost continuous manner, for which reason an interval must be chosen that yields global data. In general, noticeably short time intervals will show more specific behaviors, while very long periods will emphasize more global behaviors. In the case study presented, the traffic data generated by a system made up of nine sets of FC28 sensors coupled to the same number of ESP8266 development platforms interconnected with each other as shown in Figure 2. These connections were made through a wireless network with WiFi technology under the 802.11n standard that supplied the soil moisture data.

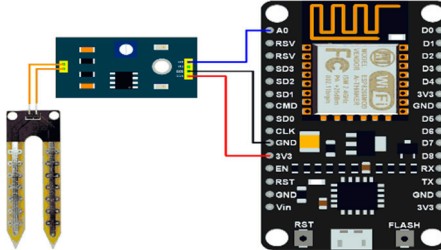

**Figure 2.** ESP8266-NodeMCU and Sensor FC-28 mounting diagram.

Figure 3 shows the implementation of a real prototype of the ESP8226 sensor FC28 set for the measurement of humidity parameters in a small Colombian Creole potato crop.

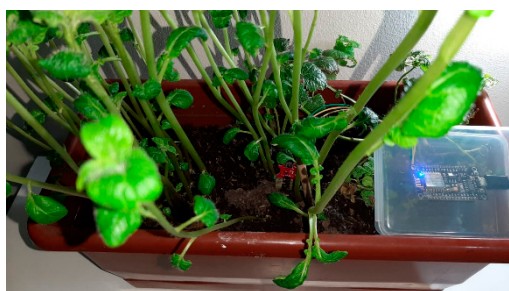

**Figure 3.** Prototype of ESP8266 and Sensor FC-28 in potato cultivation.

The distribution of the nine sensors was carried out homogeneously throughout the analyzed area of about 50 square meters. In this way, each sensor was always within a distance of no more than 50 m from the range of the WiFi gateway. The nine sensors were configured in different channels for the 2.4 GHz frequency band to avoid collisions in radio signals. In these environments with low radio frequency noise, signal losses were negligible, less than 1%. In general, all the signals measured between each sensor and the WiFi gateway had a low RSSI (Received Signal Strength Indicator) level. All measurements were below –50 dBm, which is an excellent signal level.

Some of the factors that can attenuate the quality of the transmission over WiFi signals can be: High radio frequency noise, terrain horography, thickness of the crop, or the quality of the transmitting and receiving antennas. Rain and climatic and atmospheric conditions are factors that, in general, do not affect the transmission of signals below 10 GHz. However, sometimes it has been shown that these external conditions greatly affect the quality of connection 2.4 GHz networks. For this reason,

the study of external conditions is a key factor when designing the appropriate configuration for a WSN system such as the one proposed in this study.

Due to the periodic 3-min transmissions, approximately 200 samples were taken. Once the data were organized, they were stored in GRETL as a time series [27], with a frequency of 32 samples (seasonal frequency of the series obtained by calculating the average number of samples between peaks) and any starting reference since a daily, weekly, etc. index was not followed. They were then graphically represented to observe the evolution of the variable over time. Figure 4 shows the graphic representation of the original series.

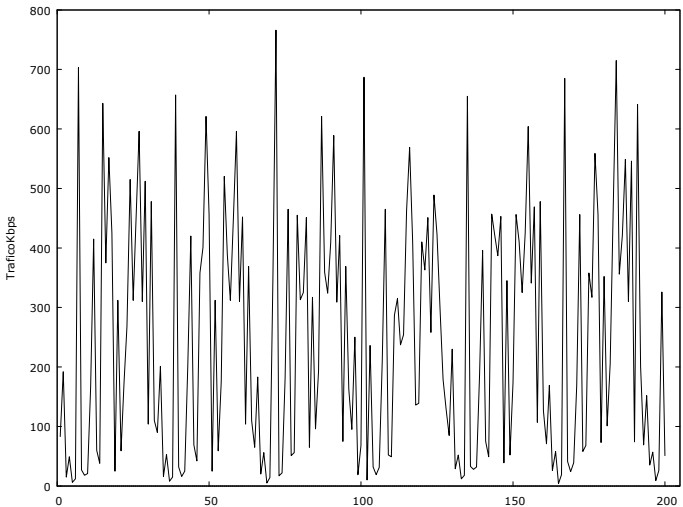

**Figure 4.** Data traffic measured in kbps for 200 s.

As can be seen, with the naked eye it is not possible to determine stationarity, since it does not present a clear trend or a defined variance. Therefore, the characteristics of the series must be analyzed in more depth.

*5.2. Analysis of Stationarity and Seasonality*

First, it was necessary to carry out an analysis to determine the stationarity and seasonality of the series using the following graphs obtained in GRETL (Figure 5).

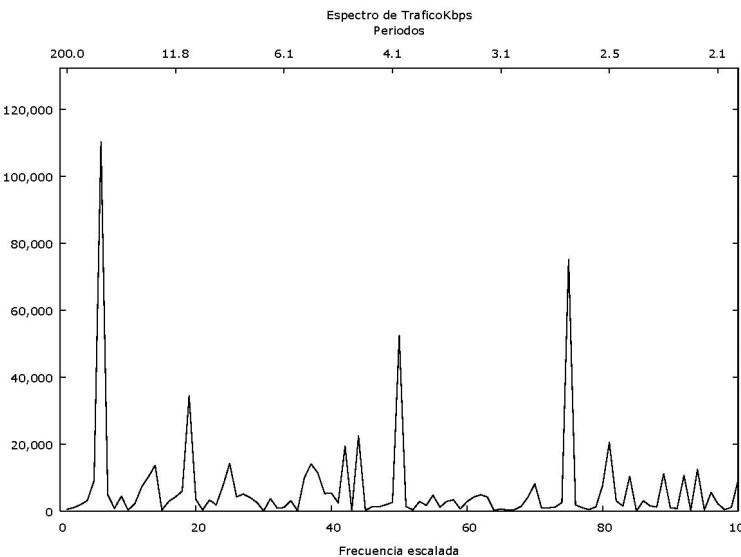

**Figure 5.** PERIODOGRAM of the original series.

The seasonal component (which is obtained based on the average number of lags between similar values) was deduced from the series periodogram, from where it can be confirmed that the series exhibited a cyclical behavior every 32 samples. The 200 analyzed samples were covered, which corresponded to omega = 0.03142, and from there the seasonal period was equivalent to 31.89.

However, it was not possible to establish whether the series was stationary since a constant mean was not obtained over time, nor was an increasing or decreasing trend. However, in addition to the trend and variance analysis, it was also possible to perform iterated differentiations to observe which one had a lower standard deviation and, thus, determine the value of "d" [28]. Table 1 shows the variance values for different degrees of differentiation (d) of the original series.

**Table 1.** Variance and standard deviation for different values of d.

| d | 0 | 1 | 2 | 3 |
|---|---|---|---|---|
| Variance ($s^2$) | 41440.7 | 70410.6 | 206997.7 | 681400.7 |
| Typical deviation (s) | 203.57 | 265.35 | 454.97 | 825.47 |

As can be seen, the variance began to increase after the second differentiation. Avoid over-differentiating the original series and eliminating valuable information that would be manifested in the autocorrelation function, since in a case of over-differentiation the autocorrelations become even more complicated to analyze, the model loses parsimony, the variance increases, and observations are lost. Therefore, it was determined that the series must be differentiated once.

The seasonality of the series can be checked by observing its ACF (Autocorrelation function) [29] and PACF (Partial autocorrelation function) functions, which are shown in Figure 6.

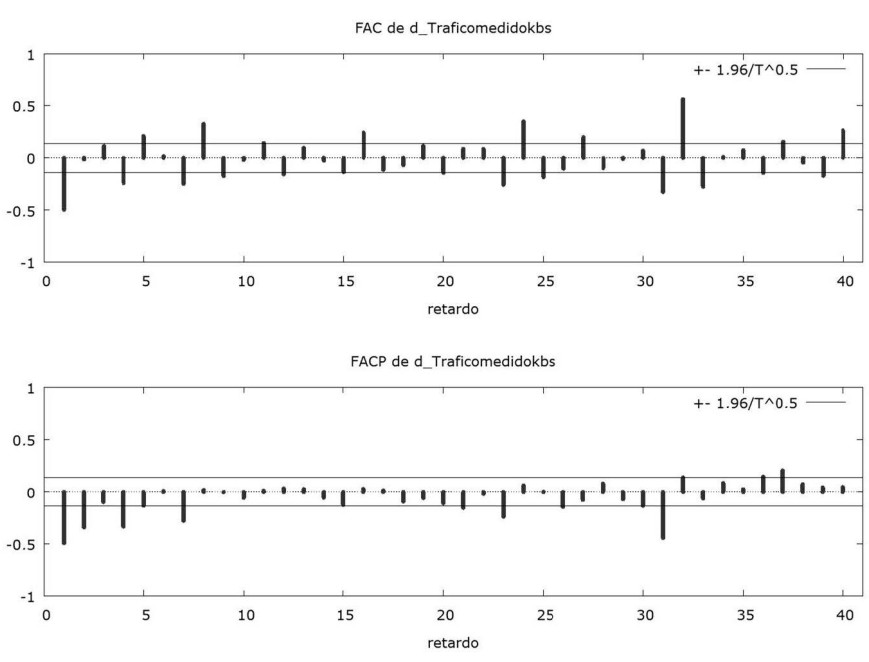

**Figure 6.** ACF (Autocorrelation function) and PACF (Partial autocorrelation function).

As can be seen, these functions yielded significant values in the delays close to 32 and other multiples of the frequency, which showed the periodic behavior of the series. Furthermore, as described in the following section, the order of the other parameters of each model was established from these graphs.

*5.3. Determination of the Model Parameters*

Continuing with the analysis of Figure 8, it was also observed that within the periods of seasonality (among the lags' multiples of the frequency) the most significant autocorrelation occurred with lags 1 and 5 and with some additional values in lags 24, 31, and 32, while the partial autocorrelation yielded a considerable value for the delay 1, 2, 4, and 31. From the previous observations, the AR (31) and MA (32) models and their combination ARMA (31,32) are obtained. However, as explained in [28], the behavior observed in the graph could also refer to an AR signature (1), since the PACF decayed relatively slowly and the ACF showed an abrupt cut in the fourth delay. This indicates that the autocorrelation with the fourth lag spread to higher lags, so another candidate to consider is the MA (4).

For the seasonal component of the losses, the analysis of the transformed series was performed by means of a differentiation of order (D) equivalent to 32. Due to this differentiation to include seasonality in the SARIMA model, the value 1 was taken for D. Both the ACF and the PACF presented significant values for seasonal lags 1 and 2. From the analysis described, candidate models with SAR (2) (Seasonal autoregressive) and SMA (2) (Seasonal moving average) parameters were obtained, which additionally (including the analysis done for the ARIMA models within the periods of seasonality) had components AR (1) and MA (4). With the above, we had the following values or ranges for the specified parameters.

From the possible combinations obtained by varying the parameters described in Table 2, 1190 ARIMA and 144 SARIMA models were obtained, many of which yielded similar results to others, taking into account that only some coefficients varied. Therefore, the range of the variables p and q was limited to take values 1 to 4 and 31 to 32 (important values in the autocorrelation function), with which 25 possible ARIMA models to be analyzed were obtained.

**Table 2.** Ranges of values for the different parameters of each model.

|   | ARIMA | | SARIMA | |
|---|---|---|---|---|
|   | **Min** | **Max** | **Min** | **Max** |
| p | 0 | 31 | 0 | 3 |
| d | 1 | 1 | 1 | 1 |
| q | 0 | 32 | 0 | 3 |
| P | N/A | N/A | 0 | 2 |
| D | N/A | N/A | 1 | 1 |
| Q | N/A | N/A | 0 | 2 |

*5.4. Estimation of the Coefficients and Validation of the Models*

To estimate the coefficients, the GRETL software tool was used, which performed the required iterations based on the specified values of the parameters p, d, q, P, D, and Q (parameters obtained from the Autocorrelation and Partial Autocorrelation functions for the analyzed time series), to obtain the coefficients of each of the models [27], in addition to yielding the calculations of the AIC function (Akaike Information Criterion) [30] with which the effectiveness of each one was validated to represent the specified series. The prediction of known future values was made through the Analysis tool of the same software, which made the automatic prediction based on the previously generated ARIMA model and the number of specified values.

**6. Results**

For the evaluation of the obtained models, the adjustment, prediction, and complexity factors were considered. The fit was measured using the Akaike Information Criterion (or AIC) [30], which measures the goodness of fit of a given model to a set of known data. This index was used as a criterion to determine the best fit for the series treated [31]. The prediction was measured by means

of the root mean square error (or RMSE) between the original data and those obtained by the model. With this measure, it was possible to compare how far the predictions were from the real data. Finally, the complexity of the models was evaluated considering the number of resulting coefficients and the computational resources they required.

*6.1. Adjustment*

A total of 190 models were analyzed, varying the parameters p, q, P, and Q. Of the 20 ARIMA models that were processed, the most significant and with the most accurate predictions were chosen. Some of these models could not be analyzed because their calculation iterations that were carried out sometimes generated values classified as infinite. In the analyzed SARIMA models, those with greater precision in behavior and prediction were taken with respect to the original series, for a total of 32 more representative models. The GRETL software calculated the AIC index when generating each model. Table 3 summarizes the main results obtained for the different models.

**Table 3.** AIC (Akaike information criteria) coefficient for various models.

| Modelo | AIC |
|--------|-----|
| ARIMA(1,1,1) | 2688.745 |
| ARIMA(1,1,31) | 2737.489 |
| ARIMA(1,1,32) | 2696.750 |
| ARIMA(2,1,24) | 2719.217 |
| ARIMA(2,1,31) | 2722.380 |
| ARIMA(2,1,32) | 2692.427 |
| ARIMA(4,1,31) | 2774.920 |
| ARIMA(4,1,32) | 2744.881 |
| ARIMA(31,1,24) | 2770.154 |
| ARIMA(31,1,31) | 2776.326 |
| ARIMA(31,1,32) | 2742.363 |
| SARIMA(1,1,1)(0,1,2)(32) | 2291.074 |
| SARIMA(1,1,1)(1,1,2)(32) | 2330.456 |
| SARIMA(1,1,32)(0,1,2)(32) | 2330.456 |
| SARIMA(1,1,32)(1,1,2)(32) | 2326.070 |
| SARIMA(2,1,32)(0,1,2)(32) | 2326.440 |
| SARIMA(2,1,32)(1,1,2)(32) | 2325.494 |
| SARIMA(32,1,32)(1,1,2)(32) | 2340.973 |

Models with a lower AIC presented a better goodness of fit and, therefore, a better representation of the data. The ARIMA model that presented the best fit by AIC was the ARIMA (1,1,1), whose mathematical description is reflected in Equation (6). It should be noted that these equations follow the notation described in the theoretical framework:

$$(1 - 0.465L)(1 - L)(Y_t) = (1 - 0.446L)a_t. \tag{6}$$

The prediction made by GRETL for this model is shown in blue in Figure 7 together with the original series (in red).

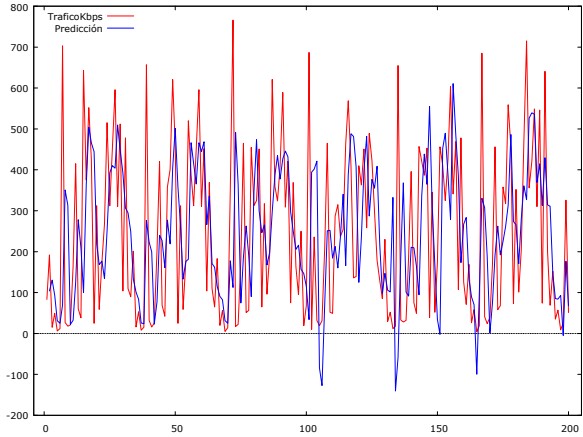

**Figure 7.** Prediction of the ARIMA model (1,1,1)**.**

The adjustment was made on a part of the original series (first 100 samples) considering that the remaining part was used to make the prediction.

On the other hand, the SARIMA model with the best AIC was the SARIMA (1,1,1) (0,1,2) (32), which is described by Equation (7).

$$(1 - 0.845L)\left(1 - L^{32}\right)(1 - L)(Y_t) = \left(1\ - 0.845L\right)\left(1\ - 0.09L - 0.904L^2\right)\left(1\ - 0.99L^{32}\right)u_t \tag{7}$$

Figure 8 shows the prediction (in blue) alongside the original series (in red).

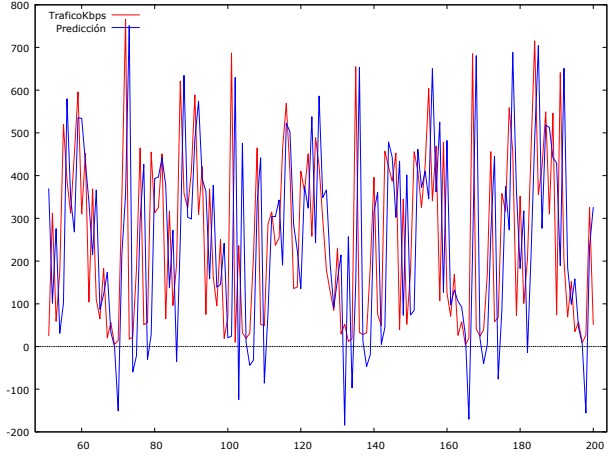

**Figure 8.** Prediction of the SARIMA model (1,1,1) (0,1,2) (32).

Although graphically the two models show similarities, mathematically the SARIMA model presented a better fit, since its AIC was 2291.074 compared to the value of 2688.427 obtained for the ARIMA model. In general, all the SARIMA models presented better AIC indices, with an average value of 2329.9 compared to the ARIMA models, which presented an average AIC of 2695.5.

In the next stage, a prediction was generated from a data set and the ability of the models to estimate the actual data was compared. As mentioned above, the models were adjusted on the first 146 samples, after which a prediction of the next 54 samples was made. Table 4 shows some of the models and the mean square error that presented their prediction against the actual values of the original series.

**Table 4.** RMSE (Root-Mean-Square Error) value for various models.

| Modelo | RMSE |
|---|---|
| ARIMA(1,1,1) | 205.38 |
| ARIMA(1,1,31) | 222.66 |
| ARIMA(1,1,32) | 178.8 |
| ARIMA(2,1,24) | 210.47 |
| ARIMA(2,1,31) | 215.46 |
| ARIMA(2,1,32) | 173.62 |
| ARIMA(4,1,31) | 245.36 |
| ARIMA(4,1,32) | 213.77 |
| ARIMA(31,1,24) | 237.69 |
| ARIMA(31,1,31) | 245.66 |
| ARIMA(31,1,32) | 212.53 |
| SARIMA(1,1,1)(0,1,2)(32) | 130.29 |
| SARIMA(1,1,1)(1,1,2)(32) | 128.56 |
| SARIMA(1,1,32)(0,1,2)(32) | 134.45 |
| SARIMA(1,1,32)(1,1,2)(32) | 133.78 |
| SARIMA(2,1,32)(0,1,2)(32) | 132.65 |
| SARIMA(2,1,32)(1,1,2)(32) | 134.12 |
| SARIMA(32,1,32)(1,1,2)(32) | 133.89 |

As can be seen, the predictions made with SARIMA models presented lower mean square errors than those obtained with higher-order ARIMA. This translates into better predictions of the real data, since the estimated values are, on average, closer to the original series.

Equation (8) describes the ARIMA model (2,1,32), which presented the best RMSE (173.62) among the ARIMA models.

$$\left(1 - 0.597L - 0.281L^2\right)(1 - L)(Y_t) = \left(1 - 0.484L + 0.417L^{32}\right)a_t \tag{8}$$

The prediction made by this model is shown in Figure 9 (in blue) together with the original series (in red).

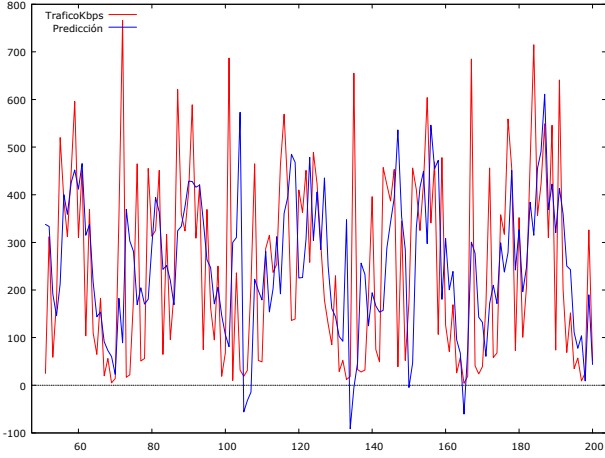

**Figure 9.** Prediction and fit for the ARIMA model (2,1,32).

Although this model estimated the random behavior of the original series in a good way, a more adjusted prediction can be observed in the ARIMA model (1,1,1). On the other hand, the SARIMA model that made the best prediction was the SARIMA (1,1,1) (1,1,2) (32), with an RMSE of 128.56 that was previously described in the Table 4. Equation (9) describes the SARIMA model mentioned in the previous sentence:

$$(1 - 0.845L)\left(1 - L^{32}\right)(1 - L)(Y_t)(1 - 0.845L)\left(1 - 0.09L - 0.904L^2\right)\left(1 - 0.99L^{32}\right)u_t \tag{9}$$

The prediction for additional data in the series was made through GRETL, considering the confidence intervals close to 95%, as can be seen in Figure 10.

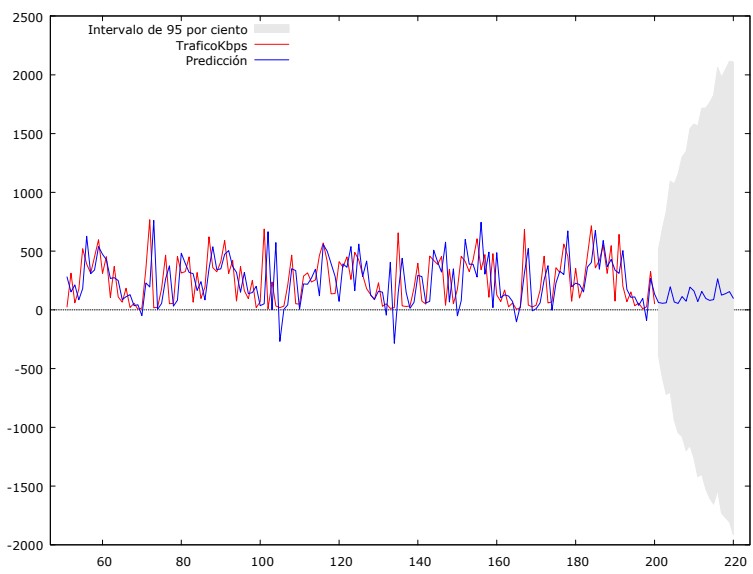

**Figure 10.** Prediction with 95% adjustment for the SARIMA model (1,1,1) (1,1,2) (32).

In the analysis of the reliability and validation of the model, the following indicators were obtained that show the appropriate adjustment for the needs raised:

(1)　Mean error: 2.9569
(2)　Root mean square error: 112.53
(3)　Mean absolute error: 65.64
(4)　Mean error percentage: –22.235
(5)　Mean absolute error percentage: 94.45
(6)　U of Theil: 0.65567
(7)　Proportion of bias, MU: 0.00019357
(8)　Disturbance ratio, UD: 0.74771

Where U, MU, and UD are measures of relative precision that compares the predicted results to the forecast results with minimal historical data.

## 7. Discussion and Conclusions

Currently, the concept and applications of the IoT (Internet of Things) are gaining more and more relevance in our world. This happens thanks to the possibility of interconnecting any object or device through the network using various communication technologies and protocols. To carry out this interconnection, the teams used various types of networks, from the best known and widely disseminated ones, such as WiFi (Wireless Fidelity), LTE (Long-Term Evolution), Bluetooth, etc. The proposed system aimed to validate a WSN environment in medium-scale crops for a specific crop

variation and in a specific area. One of the main objectives of this work was to minimize the cost of the sensorization devices, optimizing the technical characteristics that the deployed devices should have. Thanks to this information, it will be possible to generate large-scale, sensorized, and sustainable crops, as shown in Figure 11.

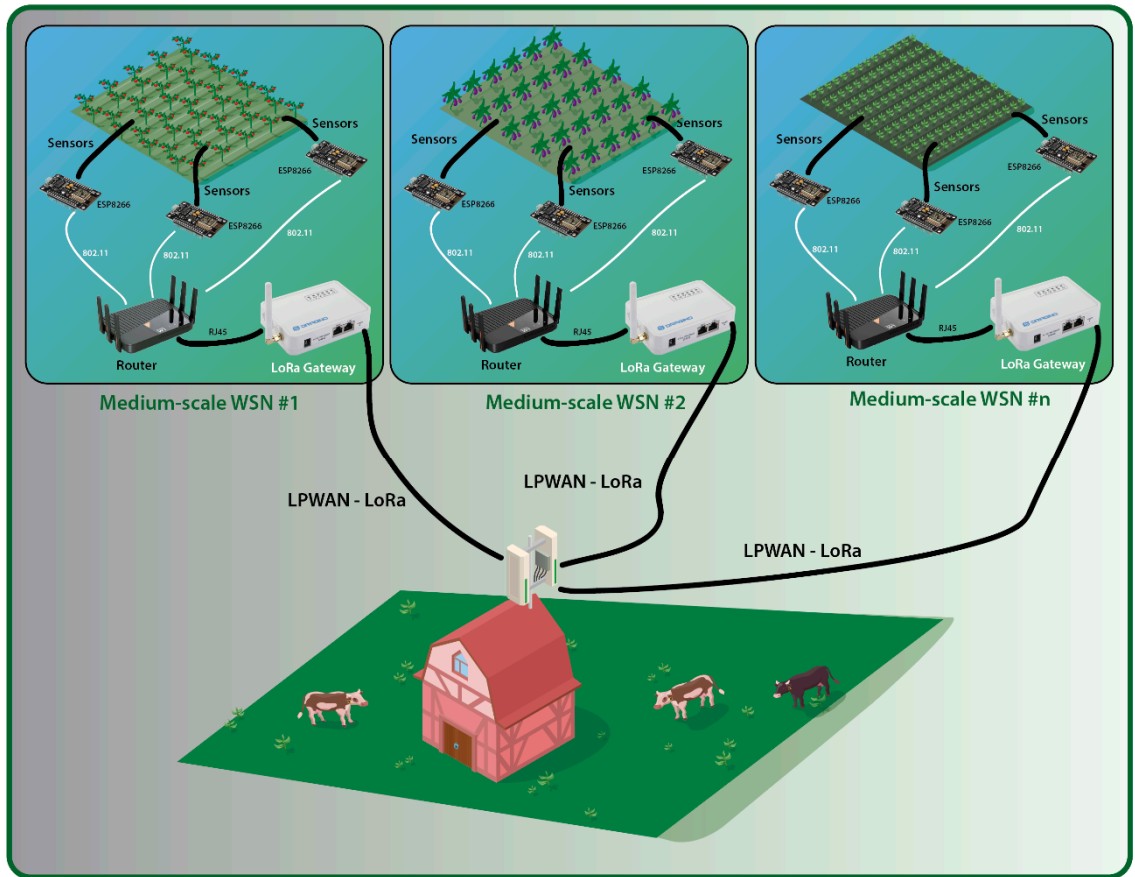

**Figure 11.** Mixed implementation of systems based on 802.11 and LPWAN wireless technology in medium and large scale. In the case of a potato crop in the Castilian plateau of the Iberian Peninsula, it would be possible to manage one hectare of crop with two medium-scale WSN (Wireless Sensor Network).

Each of these specialized communications protocols for IoT applications has the potential to become future standards with global acceptance and with significant benefits at the time of implementation. However, it is necessary to know the specifications and technical capabilities of each one of them to choose the most appropriate technology according to the environment and its application. For these reasons, the macro project that gave rise to this article proposed the creation of a model that allows the analysis and simulation of the behavior of the main performance parameters (such as energy consumption, range, and coverage; supplied bandwidth; rate of errors; and losses due to the propagation of the technologies mentioned above) in such a way that it becomes a very complete decision-making tool that contributes to research and the generation of new knowledge in this area and allows its implementation in environments such as Colombian and Spanish rural areas. Some of the parameters that the project allowed us to determine were the distance between nodes, the power of their emission antennas, or the type of IoT devices used based on their energy consumption. Specially for the benefit of the agricultural sector, forming community networks and their connection to the internet, which allow the reduction of the technology gap in this sector, contribute to improving productivity and serve as a reference for the implementation of any project involving LPWAN networks in Colombia

and the world, depending on the chemical configuration of the soil and existing crops that have a high relationship with their humidity for several meters around. In these cases, the number of sensors required to monitor the crop is less. In those soils where the chemical composition varies within a few centimeters, a higher density of sensors is necessary. Thanks to the dynamic system of the medium-scale WSN, it is possible to incorporate up to 253 sensors per WiFi gateway, the logical limit of most commercial routers. Thanks to the configuration of large-scale WSN systems, it is possible to cover large areas of land where LPWAN networks have scope.

The ARIMA models, which consist of autoregressive parameters and moving averages related to distant lags, did not capture the periodic behavior of time series with relatively large frequencies under certain traffic conditions, such as the one that occurs in the transmission of autocorrelated data and with periodic bursts of information. However, the SARIMA models allowed us to optimally represent the behavior of the traffic in a wireless network for Smart Agro environments.

With the validation of the proposed model, one of the hypotheses raised in the macro project that generated this writing was verified, showing that the necessary predictions can be made to establish the future data traffic in the wireless network where the sets of devices are linked to humidity sensors FC28—ESP8266. The simulated environment of the potato and potato crop for which the present study was carried out will allow the calculations to be made to redesign the processing capacity and the bandwidth required in the network for the technically adequate operation in a larger-scale implementation. In this way, it will be possible to make a Smart Agro project viable in the central-eastern region of Colombia and the Salamanca region in Spain in a crop of several hectares.

The measurement of the humidity parameters in the soil and the respective processing of this information constitutes a fundamental element for decision making in technician environments of smart agriculture in crops, such as potatoes and potatoes. This leads to establishing the appropriate sowing and harvesting periods to improve the productivity of these crops in rural sectors in Colombia and Spain.

Smart farms are an especially important aspect that must be addressed by systems engineering, electronic engineering, and data science. The analysis of large volumes of information using time series, big data, data mining, and other analysis methodologies allows finding patterns that allow intelligent decision making to improve and optimize different crops in the Agro sector. Thanks to the integration of the IoT and different sensors in cloud platforms, it is possible to improve many aspects related to plant growth, saving on inputs and pest control.

**Author Contributions:** The work presented in this document had the participation of a multidisciplinary team in Colombia and Spain, where several people contributed. However, it is important to highlight the contribution of those who appear as authors: A.J.L.: Contributed to the supervision and editing of the article on issues related to data analysis and data quality worked on in this study. C.A.M.A. contributed to the conceptualization and formulation of the doctoral degree work that gave rise to the study that is presented in this writing. He contributed to the formal analysis of the data worked, the development of the research, and finally, the writing and drafting of the results presented and specifically related to topics of IoT, Smart Agriculture, and communications networks. R.F.: He contributed to the Administration of the Research project and as Tutor of the doctoral thesis that gave rise to this work. He also supervised and validated the results presented in this document. Finally, I contribute to the writing and final editing of the documents that form the basis of this work. D.H.d.l.I. and V.A.S.: They contributed with their knowledge and experience to the writing, revision and editing of the final document, which was subjected on several occasions to the analysis and scrutiny of the reviewers who were assigned by the Journal. All authors have read and agreed to the published version of the manuscript.

**Funding:** This research received no external funding.

**Conflicts of Interest:** The authors declare no conflict of interest.

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
