# Peer review of "Network Traffic Modeling in a Wi-Fi System with Intelligent Soil Moisture Sensors (WSN) Using IoT Applications for Potato Crops and ARIMA and SARIMA Time Series"

_applsci, doi:10.3390/app10217702_

Round 1
Reviewer 1 Report
Please Correct!
Line 312 4.3 Methodology (Instead of Metodologia)
Line 432 Table 3 Model (instead of Modelo)
It is a good work, covering sensitive environmental issues that are related with the sustainable and wise use of water resources agriculture, using smart rural and IoT technologies.
The References need enrichment with more ARIMA and SARIMA models. The variety and the applications of such Stochastic Models, covers many issues of agriculture hydraulics, water resources quantity and quality, hydrology, etc, in many areas worldwide. So, for the enrichment of your work, please add the references below, which are releavant and up to-dated.
Suggestion: Minor Revisions
References concerning on Water Quality Management
- Sentas A. & Psilovikos A., 2010. Comparison of ARIMA and transfer function (TF) models in water temperature simulation in dam–lake Thesaurus, Eastern Macedonia, Greece. Environmental Hydraulics, Christodoulou & Stamou (eds), Vol 2, pp. 929 – 934, CRC Press – Taylor and Francis Group, 2010.
- Sentas A. & Psilovikos A., 2010. Comparison of ARIMA and transfer function (TF) models in water temperature simulation in dam–lake Thesaurus, Eastern Macedonia, Greece. Environmental Hydraulics, Christodoulou & Stamou (eds), Vol 2, pp. 929 – 934, CRC Press – Taylor and Francis Group, 2010.
- Sentas A., Psilovikos A., Psilovikos T. & Matzafleri N., 2016. Comparison of the performance of stochastic models in forecasting daily Dissolved Oxygen in the Dam – Lake Thesaurus, Greece, Desalination and Water Treatment, Vol 57, Issue 25, pp 11660-11674.
- Sentas A., Psilovikos A., Psilovikos T. & Matzafleri N., 2016. Comparison of the performance of stochastic models in forecasting daily Dissolved Oxygen in the Dam – Lake Thesaurus, Greece, Desalination and Water Treatment, Vol 57, Issue 25, pp 11660-11674.
References concerning on Agricultural Water Management
- Psilovikos A. & Elhag M., 2013. Forecasting of Remotely Sensed Daily Evapotranspiration Data over Nile Delta Region, Egypt. Water Resources Management, Vol 27, pp 4115–4130, DOI 10.1007/s11269-013-0368-2.
- Patríciade Oliveira e Lucas, Marcos Antonio Alves, Petrônio Cândido de Lima e Silva, Frederico Gadelha Guimarães, 2020. Reference evapotranspiration time series forecasting with ensemble of convolutional neural networks. Computers and Electronics in Agriculture, Vol 177, 105700, https://doi.org/10.1016/j.compag.2020.105700
- Elhag M., Gitas I., Othman A., Bahrawi J., Psilovikos A. & Al-Amri N., 2020. Time series analysis of remotely sensed water quality parameters in arid environments, Saudi Arabia. Environment Development and Sustainability, DOI: 10.1007/s10668-020-00626-z
- Zantalis F., Koulouras G., Karabetsos S. & Kandris D., 2019. A Review of Machine Learning and IoT in Smart Transportation. Future Internet, 11 (4), 94. https://doi.org/10.3390/fi11040094.
Author Response
Best regard,
Please see the attachment

Reviewer 2 Report
The article entitled “Network traffic modeling for a prototype of humidity sensors (WSN) with Smart rural and IoT applications in potato crops using ARIMA and SARIMA time series” submitted by Carlos A. Martínez Alayón, Roberto Ferro, Alfonso José López and Daniel H. de la Iglesia for publication to the “Applied Sciences” Journal of MDPI, presents the results obtained by analyzing the data traffic originated in a prototype of humidity sensors that transmit through a wireless network.
The submitted article is structured as follows:
- Section 1: Introduction
- Section 2: State of the art
- Section 3: Potato crops in Colombia and Spain
- Section 4: Theoretical framework
- Section 5: Development of the models
- Section 6: Results
- Section 7: Conclusions
Comments and Suggestions:
As it appears from studying the paper, its content is quite interesting and addresses to the readership of the Journal as it meets the subject area of applied sciences in agriculture. In particular this research work attempts to integrate smart agriculture and IoT applications in potato crops in various rural settings. Using these measurements, the data analysis was performed through the ARIMA and SARIMA time series following the Box-Jenkings methodology. GRETL free software was used to generate a teletraffic behavior prediction model in a larger scale implementation.
The methodology which is followed in the proposed approach is clearly defined and supported by adequate experiments permitting other researchers to reproduce certain aspects of this research work. The methodology analysis, as well as the results are enriched with an efficient number of adequately presented tables and figures.
Finally the paper is in general well-structured and written in appropriate and understandable English language according to the standards of the Journal. However some minor spell check might be needed.
Some specific comments and suggestions for the authors are as follows:
- Although the Introduction in Section 1 of the paper is well structured it appears to be some kind of short and generic. It is suggested that a more thorough background regarding this research aspect should be provided in order to adequately justify the research motivations as well as to substantiate the novelty of the proposed solution.
- The state of the art presented in Section 2 is suggested to include more references in order to provide a sufficient background on related research. For assisting the authors in this direction some related works are indicatively suggested: https://doi.org/10.1109/ICSITech.2016.7852645
https://doi.org/10.1109/JIOT.2018.2879579
https://doi.org/10.3390/app10030813
https://doi.org/10.3390/s20072081
- It is suggested that the authors should include a Discussion section in the manuscript. In this section the results will be discussed and interpreted in perspective of the research hypotheses. The findings and their implications should be discussed in the broadest context possible and limitations of the work highlighted. Future research directions is also suggested to be mentioned. This section may be combined with the Conclusions into one section “Discussion and Conclusions”.
Author Response
Dear reviewer:
Please see the attachment

Reviewer 3 Report
Thank you for the opportunity to review this manuscript titled “Network traffic modelling for a prototype of humidity sensors (WSN) with smart rural and IoT applications in potato crops using ARIMA and SARIMA time series.” The research is trying to access the data transmitting traffic of an indoor WIFI network in communicating with customized wireless sensors and data loggers. The manuscript has obvious flaws that need to be addressed and improved carefully before publishing.
The title of this manuscript is VERY misleading. The editor approached to me because he though this manuscript is about smart farming. Yet, the focus of its content is just investigating the data transmission of a WIFI network without demonstrating that such setup represents the “Smart rural and IoT applications in potato crops”. To be honest, WIFI network is not applicable in any rural area due to short coverage, high cost and complicate security configurations. An indoor farming system might fit the focuses but the authors need to specify the detail hardware setup including wifi router, gateway, extender, WIFI module on data logger, number of sensors, size of the building and signal coverage as comparison to an actual indoor farming system. Basically, the I would suggest changing the title to something like “Network traffic modelling of an indoor WIFI system equipped with IoT smart soil moisture sensors” without mention anything related to agriculture.
The study only used a set of 9 sensors applied toa tiny scale indoor potato cropping system whose sample size, cropping condition, sensor placement is way off to be compared to an open field farming system. The results from this study is hard to be used in any smart agriculture in a rural area. The manuscript spent almost 5 pages to describe the distribution of potato cropping in Spain and Columbia, the Potato plant, different varieties of this plant, etc., which is unnecessary and doesn’t related to the core study. Instead, the electronic engineering knowledge that is related to the transmission speed of the signal, transmission capacity of such wireless connection, and its dependence to obstacles, etc., is very limited in this manuscript. Without such knowledge, there is no physical basis to discuss the analysis results. The current statistical measures are not innovative.
The study uses ARIMA and SARIMA method for time series analysis. However, the literature review of related studies are very limited and not evident to conclude “Among the different options that exist to model some types of traffic, the statistical models stand out,”. There are MANY researches about the wireless network traffic analysis. And ARIMA and SARIMA is definitely on the popular ones and not innovative at all.
The data of this study only contains 200 samples. Considering 3 min time interval, only 10 hours data transmission is logged. This sample size is WAY small for evidently reflecting the WIFI network data transmission. In addition, I think the authors confused about the definition of “humidity” and “soil moisture”. It could be due to the name of the sensor “soil humidity sensor”. There is no humidify in soil, only moisture. Humidity only refers to the content of water in AIR. Authors need to consult a professional researcher on such terms. Moreover, soil moisture is a relative measure that cannot be exceed 100%. In Line 184, a 115.8% soil moisture was stated. This might because the sensor was not carefully calibrated to make the saturation point as 100%. Accordingly, I can see the author doesn’t have enough experience in using such sensors which makes me doubt the rest of the hardware configurations and the confidence of the data.

Round 2
Reviewer 3 Report
After reading through the answers, I think the authors did not fully understand my comments some of which still need to be addressed.
Other than changing the topic, I think description of the main purpose of this study need to be given in terms of:
- What problem does it focus? Looks like the data transferring between ESP8266 modules in an 802,11n WIFI network in an indoor system
- Why is the problem important to its application? As I understand it is a prototype of a smart agriculture for potato farming. However, is the configuration of the prototype being equivalent to the application? In other words, is the real project also using 802.11n WIFI for data collection? If not, what is the difference? And what is the associated impacts on data transferring which is the focus of this study?
- What are the uncertainties that the analysis was investigating? You collected 200 samples with 3-minutes interval. So, within this 10 hours of data collection, is the analysis mainly modelling the regular fluctuation of the network signal, or ESP8266 modules’ response effectiveness of the programmed 3-minutes data sending, or the stability of this data transferring with multiple sensors, etc. The current description only states the data collected by the router over time. What is the meaning of this information regarding to the original question that is important to the main application?
The authors added a discussion section in which a broad description was provided
“For these reasons, the macro project that gave rise to this article proposes the creation of a model that allows the analysis and simulation of the behavior of the main performance parameters, such as energy consumption, range and coverage, supplied bandwidth, rate of errors and losses due to the propagation of the technologies mentioned above, in such a way that it becomes a very complete decision-making tool that contributes to research and the generation of new knowledge in this area and allows its implementation in environments Colombian and Spanish rural areas, especially for the benefit of the agricultural sector, forming community networks and their connection to the internet that allows the reduction of the technology gap in this sector, contributes to improving productivity and serves as a reference for the implementation of any project involving LPWAN networks in Colombia and the world.”
First, you can’t put a sentence of 10 lines. Hard to read. I think this would be a good point to put at the beginning as the motivation of this study or the series of studies this one belonging to. However, it should be specifically stated that what is the narrow focus of this study. Is it the rate of errors affected by network strength only, or range and coverage?
- Innovation in this study is not well stated, the answers stated that this paring of ARIMA and ESP8266 platform is innovative. However, this is not scientifically sound. I can apply a random scratch to search for a good stock purchase without knowing if it works just caring that no one did before. The innovation should be derived on the problem this study is solving. For example, if no one have used ARIMA or SARIMA to assess the state of errors in a data transferring case. If you do want to emphasize ESP8226 module, you need to justify why this model should use ARIMA to represent its innovation.
After addressing the above concerns, I think another big concern is the representativeness of such configuration and problem-solving processes can be extended for future application.
- What is the 200 samples, 10-hour data collection, representing in the main project? In other words, if it represents the state of error, is the error reflected on the actual value and timing of sensors? Some justifications of citation or preliminary analysis are needed.
- A 32 seasonal frequency was stated in the analysis which is an important parameter for further ARIMA and SARIMA methods. How was this number determined? Manually counting the peaks? How the peaks are accounted, with an arbitrary threshold, or automatically determined with a dynamic process? If there are multiple season frequencies, e.g. daily fluctuation and seasonal temperature shifts, presented in a real project, how this method will be affected? Obviously, the 10-hour data collection in this case can’t cover the frequencies longer than half-day. But what if you have data more than a month, and have daily, weekly or monthly frequency presented? How would you identify these frequencies before using ARIMA or SARIMA to model the residue stationary series?
In addition, it is okay to describe briefly, 2 paragraphs or 1 page as maximum, about the main project this study belonging to. This manuscript only has 19 pages of text. But 5 of 19 were talking about potato. Does the analysis or application only valuable on potato in Columbia and Spain? You can refer to another publication that is specifically describing the overall project and science of potato instead of bringing a 5-page distraction from your method. Length doesn’t count for value for a journal paper.
Author Response
Cordial and respectful greeting. Please see the attachment

Round 3
Reviewer 3 Report
Please see the highlighted responses to your answers

Author Response

(The authors gave the same response as above.)
